# Degradation Characteristics of Phosphorus in Phytoplankton-Derived Particulate Organic Matter and Its Effects on the Growth of Phosphorus-Deficient *Microcystis aeruginosa* in Lake Taihu

**DOI:** 10.3390/ijerph16122155

**Published:** 2019-06-18

**Authors:** Ming Kong, Jianying Chao, Wei Han, Chun Ye, Chun-Hua Li, Wei Tian

**Affiliations:** 1Nanjing Institute of Environmental Sciences, Ministry of Ecology and Environment, No. 8 Jiang Wang Miao Street, Nanjing 210042, China; kongming@nies.org (M.K.); tw79210@163.com (W.T.); 2Sino-Japan friendship Center for Environmental Protection, Beijing 100029, China; hanwei_2002@126.com; 3Chinese Research Academy of Environmental Sciences, National Engineering Laboratory for Lake Pollution Control and Ecological Restoration, Beijing 100012, China; yechbj@163.com

**Keywords:** Phytoplankton, particulate organic matter, degradation, phosphorus, *Microcystis aeruginosa*, Lake Taihu

## Abstract

To illustrate the contribution of phytoplankton-derived particulate organic matter (PPOM) to endogenous phosphorus (P) cycling and its effects on cyanobacteria blooms, PPOM characteristics, the degradation mechanism, and the growth of P-deficient *Microcystis aeruginosa* were studied in Lake Taihu. Results showed that PPOM is the most important P pool in the water column during cyanobacteria bloom, accounting for more than 80% of the total P (TP) in the water. During PPOM degradation, the particulate orthophosphate (Ortho-P) is the main species of P release from PPOM in the early degradation stage. The variations of polyphosphate (Poly-P) and phosphodiesters (Diester-P) contents were most significant, which were degraded completely within four days and eight days. Cell density and growth rate of *M. aeruginosa* using PPOM as P source were similar to those growing on Na_2_HPO_4_. The above results show that P in PPOM can be converted into available P by degradation, thus promoting the growth of *M. aeruginosa*. Therefore, the contribution of P release from PPOM degradation needs to be paid attention to in lake eutrophication control in the future.

## 1. Introduction

Lake eutrophication has become one of the major ecological and environmental problems worldwide and can lead to the frequent occurrence of cyanobacterial blooms [1,2]. Phosphorus (P) is an indispensable element for phytoplankton growth in freshwater systems and a determinant factor of eutrophication [3]. Once the inputs of exogenous P are effectively controlled, internal P recycling is the most important source of P for primary production and maintaining the trophic status and cyanobacterial blooms [4]. 

Particulate organic matter (POM) is an important component of aquatic ecosystem and a nutrient carrier [5]. In lake water ecosystems, POM degradation is accompanied by P cycling [6]. In the water, POM includes exogenous imported POM and endogenous POM. Specifically, exogenous imported POM mainly comes from river input [7], while endogenous POM mainly comes from phytoplankton, zooplankton, and their metabolites, as well as microorganisms [8]. Another important POM source in the water is through resuspension from the sediments [9,10]. The source and composition of POM in different types of water bodies are quite different. Generally speaking, there is more endogenous POM in eutrophic water, and resuspended sediment is the main POM source in shallow water ecosystems, especially in eutrophic shallow lake water, where complex hydrodynamic and biological factors complicate the source, contents, composition, and bioavailability of POM [11,12,13].

In eutrophic lakes, phytoplankton-derived particulate organic matter (PPOM) is an important component of water POM, mainly containing phytoplankton and its residues after death [14,15]. A large number of microorganisms can be attracted or reproduced on the surface of PPOM to carry out strong bio-metabolic activities [16,17]. The more easily decomposed PPOM can rapidly degrade through a series of redox and phosphatase hydrolysis processes, releasing various species of P to the surrounding water body [18]. Part of P entering the water body is dissolved in water and can be directly absorbed and utilized by algae and other organisms [19]; part of P enters the sediment with the deposition of particulate matter and then suspends under the disturbance of wind, waves, and organisms, re-entering the overlying water body and participating in the P cycle [20]. Therefore, the degradation process of PPOM may play an important role in affecting P migration and distribution.

At present, there are many studies on POM in the ocean [21,22], and it has been recognized that the metabolic characteristics of POM in water bodies are closely related to their nutrient status and hydrodynamic conditions [6]. However, less research has focused on the characteristics of POM in shallow eutrophic lakes, which are completely different from those in oceans and deep lakes. *Microcystis* is one of the dominant genera in phytoplankton of eutrophic lakes [23]. Various species of P in water body have different nutritional supply abilities to the growth of *Microcystis*. Soluble reactive P (SRP) in water body can be directly utilized by *Microcystis*. When the supply of SRP in water body is insufficient, *Microcystis* can hydrolyze organic P into inorganic P by releasing phosphatase to maintain its growth. However, the effect of P degradation in PPOM on *Microcystis* growth is still unclear, which limits our understanding of the bioavailability of PPOM. The main purposes of this study are (1) to clarify the variation of particulate P composition during PPOM degradation in water of shallow eutrophic lake, and (2) to reveal PPOM bioavailability and its effects on the growth of phosphorus-deficient *Microcystis*.

## 2. Materials and Method

### 2.1. Study Area

Lake Taihu, located in the Yangtze River Delta, is the second largest shallow freshwater lake in China. It has a water surface area of 2427.8 km^2^ and an average water depth of 1.9 m [24]. It is an important resource for drinking water, shipping, freshwater aquaculture, and farming. Recent toxic cyanobacterial blooms caused by excessive human nutrient loads have focused attention on controlling blooms and restoring the lake to acceptable water quality and nutrient conditions [1,25,26]. Water samples for simulated experiments were collected in Meiliang Bay (120°12′49.67″ E, 31°29′09.99″ N), which is located in the northern part of Lake Taihu (Figure 1). Meiliang Bay has been eutrophicated and suffered from harmful algal blooms during the past decades, where *Microcystis aeruginosa* (*M. aeruginosa*) is one of the dominant species [27]. The excessive growth of toxic *M. aeruginosa* greatly deteriorated the water quality, damaged the natural functions of lake system, and even threatened the drinking water resources.

### 2.2. Experimental Design

#### 2.2.1. Simulation Experiment on Degradation of PPOM

Water samples were taken in Meiliang Bay of Taihu Lake in August 2014; plastic buckets (80 L) were used in the simulation experiment. Three parallel groups were set in the experiment. The experiment device was put into a pool in the glass room for constant temperature (25 ± 1 °C) and natural light conditions. Small electric agitators were placed on top of the plastic bucket to avoid particle settling. Water samples (0.5 L) were taken on days 0, 1, 2, 4, 5, 6, 7, 8, 11, 14, and 20, which were used for the determination of physical and chemical indexes. The samples were filtered by a GF/F membrane to collect PPOM.

#### 2.2.2. Effects of PPOM Degradation on the Growth of Microcystis Aeruginosa

Water samples (50 L) were taken at the same time and place as we mentioned in the Section 2.2.1., and PPOM was collected by plankton net (mesh size 64 μm). Then, the samples were prepared by freeze drying, grinding, and being passed through 63 μm mesh sieves. In this experiment, five species of P, including K_2_HPO_4_ (orthophosphate: Ortho-P), G-6-P (monophosphate: Mono-P), lecithin (phosphodiester: Diester-P), Na_5_P_3_O_10_ (pyrophosphate/polyphosphate: Pyro-P/Poly-P) and PPOM were selected as P sources, which define group 1 (G1), group 2 (G2), group 3 (G3), group 4 (G4), and group 5 (G5), respectively. *M. aeruginosa* was obtained from Nanjing Institute of Geography and Lakes, Chinese Academy of Sciences. *M. aeruginosa* in the S phase were centrifuged at 5000 *g* for 10 min, washed with P-free medium, and then centrifuged again at 5000 *g* for 10 min. After repeating the experiment three times, *M. aeruginosa* was transplanted into P-free medium for 15 days and then into sterilized medium with different P sources. The initial cyanobacteria density was (1.9 ± 0.1) × 10^6^ cells/mL; culture temperature was controlled at constant temperature (25 ± 1 °C). The ratio of illumination time to darkness time was 12 h:12 h, and illumination intensity was 4000 lx. Water samples were collected and measured every two days (0, 2nd, 4th, 6th, 8th, 10th, 12th, 14th, 16th), three parallel samples were collected each time.

### 2.3. Sample Analysis

#### 2.3.1. Physical and Chemical Index Analysis

Water temperature was measured by thermometer. Turbidity of water was analyzed with a turbidimeter (2100Q, HACH, Loveland, CO, USA). Dissolved oxygen (DO) was measured with a dissolved oxygen meter (LDOTM, HACH, Loveland, CO, USA). pH was measured using a pH meter (HQ11d, HACH, Loveland, CO, USA). Suspended particulate matter (SPM) was obtained by weighing after drying (105 °C for 4 h). POM was calculated from the loss on ignition (450 °C for 4 h) [28]. 

For the determination of particulate organic carbon (POC), the prepared dry particulate powder samples were fumigated with concentrated hydrochloric acid for 12 h in an airtight dryer to remove inorganic carbon [29]. After acidification, POC was determined with an elementary element analyzer (FlashSmart, ThermoFisher, Germany). Total P (TP) concentration in water was analyzed at 700 nm by the colorimetric technique after 30 min of autoclave mediated digestion (120 °C with K_2_S_2_O_8_ and NaOH) of unfiltered and filtered samples, respectively. For total particulate P (TPP) concentrations in SPM, the Nuclepore filter samples were wetted with 0.5 M MgCl_2_ solution and heated in an oven at 95 °C until dry, followed by ashing in a furnace at 550 °C for 2 h to decompose organic P compounds. The residue was extracted using 1 M HCl solution at room temperature (25 °C) for 24 h [30]. After neutralization and dilution, both TPP and particulate inorganic P (PIP) extractions were analyzed, and the concentration of particulate organic P (POP) was then calculated from the difference between TPP and PIP. Chlorophyll *a* (Chl *a*) was measured spectrophotometrically at 750 and 665 nm after extraction of phytoplankton biomass using 90% hot ethanol [31]. The above parameters were analyzed in three parallel samples.

#### 2.3.2. Molecular Particulate P Analysis by ^31^P Nuclear Magnetic Resonance

Particulate P (PP) samples were extracted in 50 mL of acid-washed falcon tubes with 15 mL of 0.5 M NaOH and 0.1 M EDTA for 16 h at room temperature on a shaking plate, followed by centrifugation at 2000 rpm for 20 min [32]. NaOH and EDTA extraction for PP were concentrated to approximately 0.5 mL in a rotary vacuum evaporator at 28 °C for solution ^31^P nuclear magnetic resonance (^31^P NMR) spectroscopy analysis [33]. 

Before the ^31^P NMR analysis, the extracts were transferred into a 5-mm NMR tube and D_2_O was added into the supernatant to reach 10% proportion for signal lock. The ^31^P NMR spectra were measured at 161.84 MHz on a Bruker AV400 spectrometer (Bruker, Billerica, MA, USA) equipped with a 5-mm broadband probe, using a 90° pulse, relaxation delay 2 s, and acquisition time 0.5 s, obtaining around 19,000 transients. 

Chemical shifts were recorded relative to 85% H_3_PO_4_ via the signal lock [34]. Peak assignments were made using a ^31^P NMR chemical shift of Ortho-P (6−7 ppm), Mono-P (4−6 ppm), Diester-P (0−3 ppm), Pyro-P (−3.5 to −4.5 ppm), Poly-P (−17 to −19 ppm), and phosphonate (18−20 ppm). Peak area was calculated by integration and the spectra plotted with a line broadening of 3 Hz [35]. 

#### 2.3.3. Alkaline Phosphatase Activity in Particulate Matter and Water

Alkaline phosphatase activity (APA) was determined with p-NPP as the reaction substrate. A 2-mL water sample was taken, and 1 mL Tris (0.1 mol·L^−1^) and 2 mL p-NPP (0.001 mol·L^−1^) were added and cultured for 6 h under dark conditions at 30 °C. Finally, 0.5 mL NaOH (0.1 mol·L^−1^) was added to terminate the reaction process. The absorbance was measured at 410 nm by spectrophotometer after 10 min centrifugation at 5000 rpm/min. The standard curve of NPP at 410 nm wavelength was established to calculate the concentration of NPP generated from hydrolysis, so that APA could be calculated [36]. The particulate matter of APA could be obtained by the subtraction of filtered water from raw water. Enzyme hydrolyzable P (EHP) could be used as a proxy of bioavailable P in water, taking 100 mL water sample, 1 mL Tris (1.0 mol·L^−1^) buffer solution, and 5 mL chloroform into a sterilized glass bottle. Samples were incubated at 30 °C for 5 days, and P content was determined by spectrophotometry [37].

### 2.4. Statistical Analysis

The experimental data were analyzed using SPSS 19.0 for Windows (IBM, Armonk, NY, USA) and OriginPro 8.5 (OriginLab, Northampton, MA, USA). All analyses were performed using standard procedures in Microsoft Excel. 

## 3. Results

### 3.1. General Characteristics of Water and PPOM

The general characteristics of water and PPOM are shown in Table 1; the temperature of the culture system was kept consistent at about 30 °C. DO ranged from 3.42 to 7.12 mg·L^−1^, which first showed a decreasing trend and then increased. On the 3rd day of the experiment, DO reached the minimum value, then slowly increased. At the end of the experiment, DO returned to its starting value. The pH ranged from 7.58 to 7.86, and also first showed a decreasing trend and then increased. The initial suspended solids (SS) content was 135.32 mg·L^−1^ and reached the minimum (100.89 mg·L^−1^) on the 6th day after the start of the experiment. After that, the SS concentration increased significantly, with a maximum (142.75 mg·L^−1^) nearly reaching the initial concentration. Similar to the SS trend, the initial POM content was 130.12 mg·L^−1^, then it decreased and reached the lowest value (89.33 mg·L^−1^) on the 6th day. After that, it showed a clear upward trend. Chl *a* concentration showed a rapid initial decline and decreased to 461.53 μg·L^−1^ on the 6th day. Afterwards, Chl *a* basically stayed at around 400−500 μg·L^−1^. The POC concentration increased significantly on the first day of the experiment, from 56.43 to 74.05 mg·L^−1^. After that, the concentration remained at around 60 mg·L^−1^. The POP concentration showed a significant decrease during the first two days, from the initial value of 0.97 to 0.56 mg·L^−1^; after that, no significant changes were observed. Similar to the POP trend, TPP ranged from the initial value of 0.99 to 0.59 mg·L^−1^ during the first two days, and then declined slightly. TP showed the same trend with TPP and POP, and TPP accounts for most of TP (80.3–90.8%). 

### 3.2. Degradation Characteristics of PPOM

#### 3.2.1. Analysis of P Species in PPOM by ^31^P NMR

The composition change of granular P is shown in Figure 2 and Table 2; at the beginning of the experiment, five P components were detected in PPOM. Ortho-P accounted for the highest proportion of TPP with 46.68%, followed by Mono-P (38.75 %). Organic P (P_o_, including Mono-P and Diester-P) accounted for 45.41% of TPP; Inorganic P (P_i_, including Ortho-P, Pyro-P and Poly-P) accounted for 54.58% of TPP; Biogenic P (Bio-P, including Mono-P, Diester-P, Pyro-P and Poly-P) accounted for 53.31% of TPP. After the start of the experiment, Ortho-P showed a significant decrease, reaching the lowest value of 25.63% on the 1st day, while Bio-P showed a significant increase, and increased to 74.36% on the 1st day; then P_i_ began to rise slowly and Bio-P showed a slow decline. By the 8th day, Bio-P reached the minimum (51.76%), and then began to rise again, reaching a maximum of 74.29% on the 14th day. During culture, the high peak of Pyro-P + Poly-P content appeared twice, and the appearance time was consistent with Bio-P. The first high peak appeared on the 2nd day, and the second highest peak appeared on the 14th day, accounting for 14.76% and 31.40% of the TPP, respectively.

#### 3.2.2. Analysis of P Species by Enzymatic Hydrolysis

The variation of EHP concentration in PPOM is shown in Figure 3a. The EHP concentration in the particles at the beginning of the experiment ranged from 0.401 to 0.584 mg·L^−1^ in the first two days, and it began to decrease slowly on the 3rd day. According to the EHP: TPP ratio in PPOM, the ratio of particulate enzymatic P to total particulate P is 60%. This ratio rises rapidly to about 90% after one day, and remains at 90% for the next eight days. From the 9th day, the particulate EHP continues to decline, and the rate of decline is much higher than that of TPP. As shown in Figure 3b, the APA trend is similar to that of EHP. The overall trend is as follows: rapid rise, rapid decline, and finally a stable period. APA concentration reached a high peak on the first day (0.022 mmol·L^−1^), then rapidly decreased to 0.009 mmol·L^−1^ on the 2nd day followed by no significant change from day 3 to 14. Finally, APA showed a slowly decreasing trend.

### 3.3. Cell density Changes of M. Aeruginosa in Culture Systems with Different P Sources

*M. aeruginosa* cell density and growth rate changes are shown in Figure 4 and Figure 5. *M. aeruginosa* growth rate could be promoted by five kinds of media, but the growth rate was significantly different. G1 had the highest density of *M. aeruginosa* (2.3 × 10^7^ cells·mL^−1^), followed by G3 and G5 with 2.0 × 10^7^ and 1.9 × 10^7^ cells·mL^−1^, respectively. *M. aeruginosa* density increased very slowly in G2 and G4, and was one order of magnitude lower than in G1, G3, and G5. As shown in Figure 5, *M. aeruginosa* growth rate in in G5 was similar to that in G1 and G3, which was significantly higher than that in G2 and G4. As shown in Figure 6, the APA continued to rise after the start of the experiment in G5, and was significantly higher than that in the other groups, reaching a peak of 0.015 mmol·L^−1^·h^−1^ at the 14th day of the experiment. In G1, G2, and G4, APA always remained at a low level of about 0.002 mmol·L^−1^·h^−1^. In G3, APA maintained a fluctuation between 0.003 and 0.004 mmol·L^−1^·h^−1^ after the 4th day.

## 4. Discussion

### 4.1. Physicochemical Properties of PPOM

This experiment simulated the cyanobacteria degradation process during cyanobacteria accumulation in Taihu Lake. The initial concentration of Chl *a* in the culture system was about 877.45 μg·L^−1^, which was common in the cyanobacteria accumulation area of Taihu Lake Bay [38,39]. The initial content of POM in water was about 130.12 mg·L^−1^, which is similar to SS content in Meiliang Bay of Taihu Lake during summer [40] and much higher than that of Jiaozhou Bay in summer (12–17.4 mg·L^−1^) [41]. The ratio of initial POM to SS was between 85.2% and 90.6%, which was slightly higher than that in Jiaozhou Bay (58.73–72.1%) [39] and much higher than the average level of 7% in Taihu Lake [40].

Studies have used the POC: Chl *a* value to characterize the proportion of fresh cyanobacteria and cyanobacteria degradation residues in particulate matter [41,42]. It is generally believed that when the POC: Chl *a* value is less than 200, it indicates that fresh living cyanobacteria are dominant in the particles. Contrastingly, if the POC: Chl *a* value is greater than 200, dead cyanobacteria and their degraded residues are dominant in POM [43]. In this study, POC: Chl *a* averaged 70 at the beginning of the experiment and then increased, but did not exceed 200 during the experiment (Figure 7). During the process of bloom and phytoplankton proliferation, PPOM (with low POC: Chl *a* value) derived from algal growth was continuously produced.

### 4.2. Degradation Characteristics of P in PPOM

#### 4.2.1. Variation Characteristics of Organic P in PPOM

The experimental data showed that Ortho-P in PPOM decreased nearly by 60% in 12 h. This process may be caused by the adsorption of Ortho-P on PPOM. Mono-P was basically maintained at more than 30% of TPP throughout the experiment, while Diester-P decreased to 0 on the 8th day, indicating that Mono-P is more stable than Diester-P. Previous studies have also shown that the decomposition rate of Diester-P is higher than that of Mono-P [44]. In this study, Diester-P ratio first decreased and then increased. The lowest value appeared on the 8th day, which may represent the degradation of diester in PPOM. After the growth and death of cyanobacteria reached a balance, new synthesis of organic P occurred. 

Poly-P is more unstable than other species of P, and its concentration ratio dropped rapidly in the first two days of the experiment and then gradually increased. The level of Poly-P may be related to the growth stage of cyanobacteria and bacteria. In the early stage of the experiment, a large number of cyanobacteria died and Poly-P degraded. In the later stage of the experiment, the growth and decay of cyanobacteria reached a balance and cyanobacteria changed from a degradation process (death) at the beginning of the experiment to a balance between growth and degradation. Poly-P re-accumulated in the cyanobacteria due to the rise of phosphate in the water. 

Similar to Poly-P, Pyro-P also showed a trend of initial decrease followed by an increase in our study. Some studies indicated that Pyro-P was a degradation product of Poly-P under the action of microbial metabolism; Poly-P and Pyro-P are both potential bioavailable phosphorus, which are actively involved in the P circulation in water and are closely related to the nutrient level of lakes. [18,45]. 

#### 4.2.2. Variation Characteristics of Enzymatically Hydrolysable P in PPOM

Phosphatase hydrolysis is an important participant and driving force in P biogeochemical cycle in lakes [46]. EHP is a potential bioavailable P in particulate matter, which can be used by phytoplankton after hydrolysis by phosphatase [47]. The results showed that EHP in PPOM accounted for more than 60% of TPP (Figure 3); this indicated that the P in PPOM had high bioavailability. The contents of EHP and APA showed the same trend during the whole PPOM degradation process (increased rapidly at first, then decreased rapidly, and finally decreased slowly). During the degradation of PPOM, the particulate Ortho-P, which is the main species of P released from PPOM in the early stage of cyanobacteria degradation, is rapidly released into the water. This process may be due to the release of adsorbed P in PPOM. Thereafter, the rapid enzymatic hydrolysis of organic P was dominant, and the degradation ratio reached or approached 100% within two days. 

In eutrophic lakes, the increase of organic matter will lead to the increase of phosphatase activity [48]. When blooms occur, the residues of plankton can accumulate in suspended particulate matter after the plankton death, and POM can also increase. The increase of phosphatase activity promotes EHP hydrolysis in PPOM. At the same time, the accumulation of organic matter can promote the increase of EHP [49]. EHP recycling (enzymatic hydrolysis of orthophosphate) can maintain or even aggravate lake eutrophication. Therefore, water eutrophication is related to EHP content and transformation of particulate matter.

### 4.3. Effect of PPOM on the Growth of M. Aeruginosa

From the experimental results, there was no significant difference in the growth rate and density of *M. aeruginosa* among G1, G5, and G2, which indicated that G5 could effectively promote the growth of cyanobacteria. The average APA in the G5 group was 0.0095 mmol·L^−1^·h^−1^, which was significantly higher than that in other groups. This indicated that *M. aeruginosa* could utilize macromolecule organic P by releasing alkaline phosphatase, but alkaline phosphatase had no effect on small molecule organic P and inorganic P. This conclusion was similar to that in related studies [50].

It is generally believed that *M. aeruginosa* can directly absorb Ortho-P which is the main P species. When the concentration of Ortho-P in the environment decreases, *M. aeruginosa* will secrete extracellular phosphatase to degrade the organic P and release phosphate for cyanobacteria absorption and utilization, indicating that PPOM can be provide a source of P for the growth of *M. aeruginosa*. Moreover, the growth rate of *M. aeruginosa* (0.148 d^−1^) is close to that of G1 (0.156 d^−1^), indicating that PPOM has high bioavailability. During algal blooms, phytoplankton growth causes a large demand for P, and the contribution of PPOM to the availability of P in lakes should not be ignored. Our results support that the biogeochemical cycle of P in PPOM has an important impact on the water quality protection of lakes when the external P input is effectively controlled.

## 5. Conclusions

In this study, the degradation characteristics of phosphorus in phytoplankton-derived particulate organic matter and its effects on the growth of phosphorus-deficient *Microcystis aeruginosa* in Lake Taihu have been analyzed. During the degradation of PPOM, particulate Ortho-P, which is the main P species of PPOM, is rapidly released into the water body in the early degradation stage. The rapid enzymatic hydrolysis is dominant, and its degradation ratio of Poly-P and Diester-P reach or approach 100% within eight days. The effect of PPOM as a P source on the growth of *Microcystis* is similar to that of Na_2_HPO_4_. The cell density and growth rate of *Microcystis* with PPOM as a P source were similar to those of *Microcystis* having Ortho-P as P source, indicating that PPOM has high bioavailability and that P released during the degradation process can effectively promote the growth of *Microcystis*. Therefore, the contribution of P release from PPOM degradation needs to be paid attention to in lake eutrophication control in the future, and more field studies are needed to further confirm it.

## Figures and Tables

**Figure 1 ijerph-16-02155-f001:**
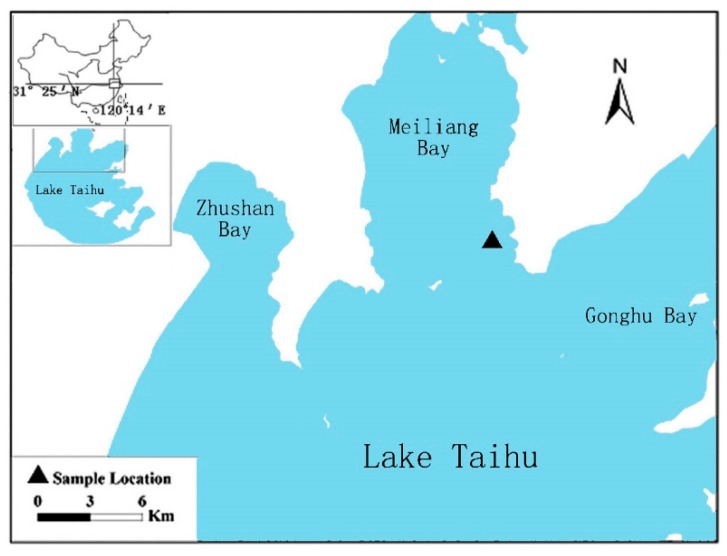
Location of Lake Taihu, China and ▲ indicates the location of the sample collection point.

**Figure 2 ijerph-16-02155-f002:**
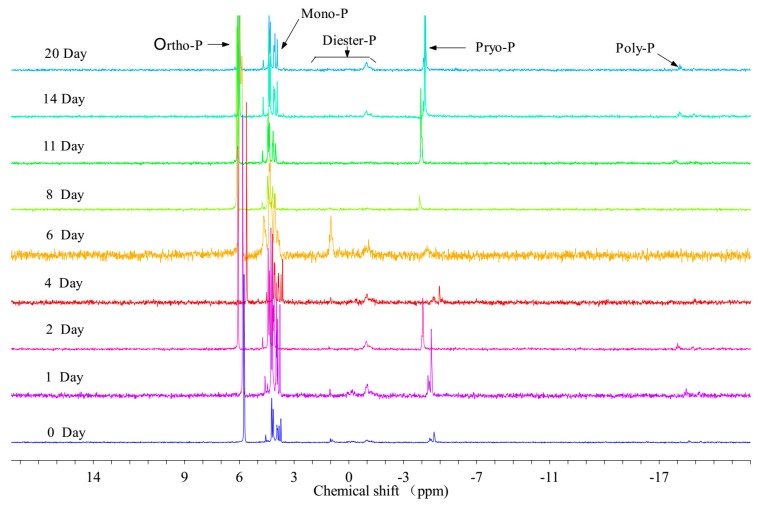
^31^P NMR spectra of PPOM (Water samples were taken and measured on days 0, 1, 2, 4, 5, 6, 7, 8, 11, 14, and 20).

**Figure 3 ijerph-16-02155-f003:**
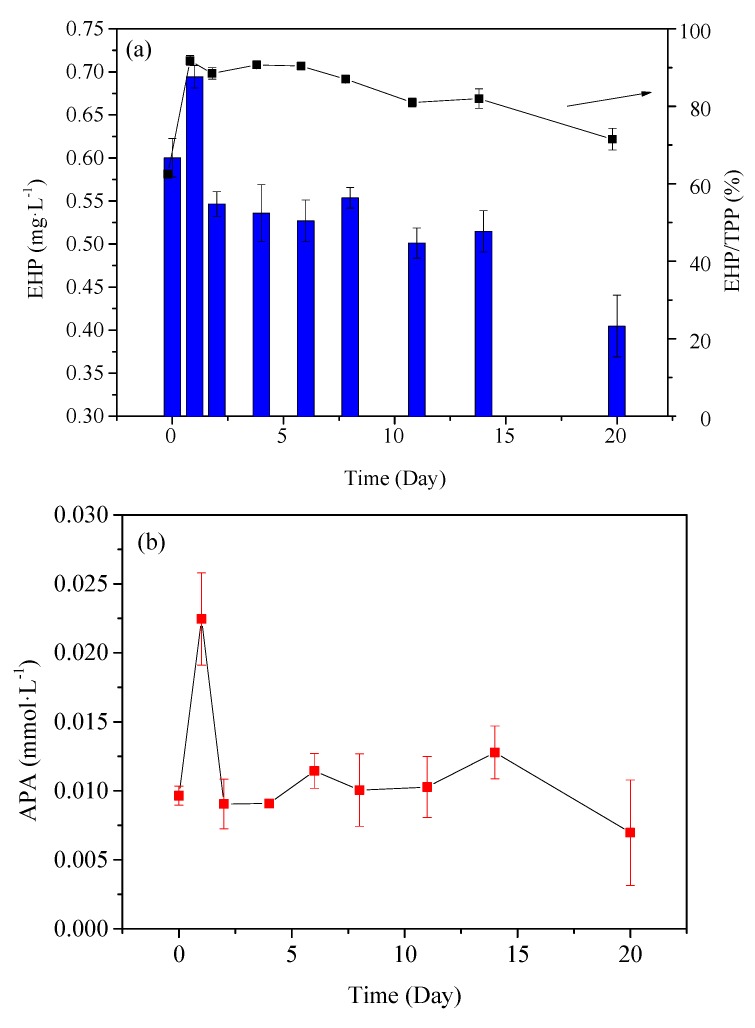
The content variation of Enzyme hydrolyzable P (EHP) and Alkaline phosphatase activity (APA) in PPOM. (**a**) The content variation of EHP and the ratio of EHP to total Particulate phosphorus (TPP), (**b**) the content variation of APA.

**Figure 4 ijerph-16-02155-f004:**
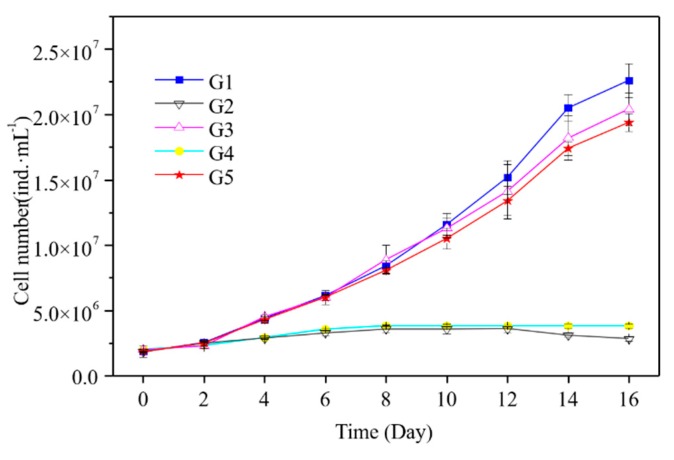
Variation of cell density of *M. aeruginosa* in culture systems with different P source. G1 (group 1): K_2_HPO_4_ (orthophosphate: Ortho-P); G2 (group 2): G-6-P (monophosphate: Mono-P); G3 (group 3): lecithin (phosphodiester: Diester-P); G4 (group 4): Na_5_P_3_O_10_ (pyrophosphate/polyphosphate: Pyro-P/Poly-P); G5 (group 5): PPOM.

**Figure 5 ijerph-16-02155-f005:**
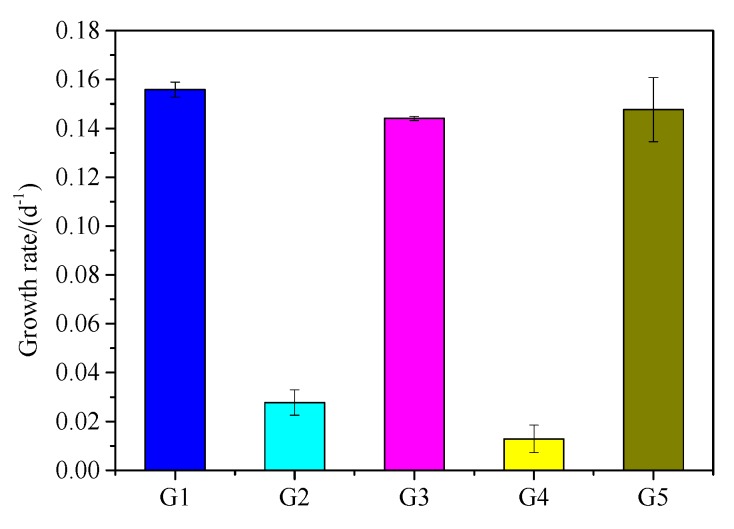
Growth rate of *M. aeruginosa* in culture systems with different P source.

**Figure 6 ijerph-16-02155-f006:**
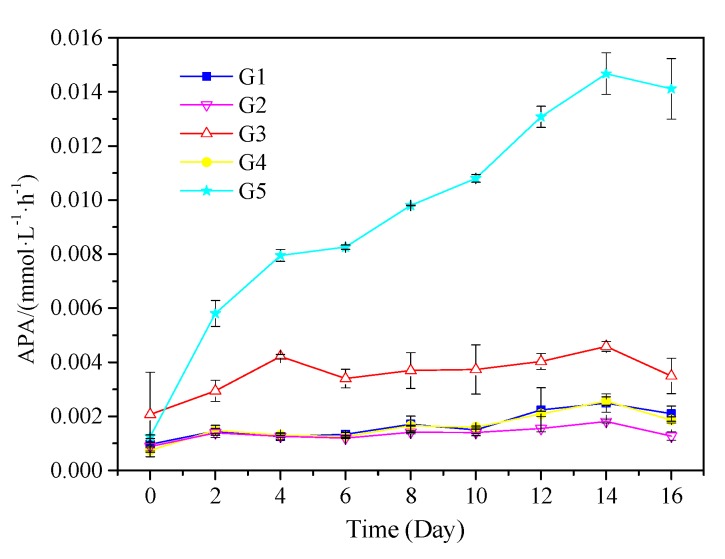
Changes of APA of *M. aeruginosa* in culture systems with different P source.

**Figure 7 ijerph-16-02155-f007:**
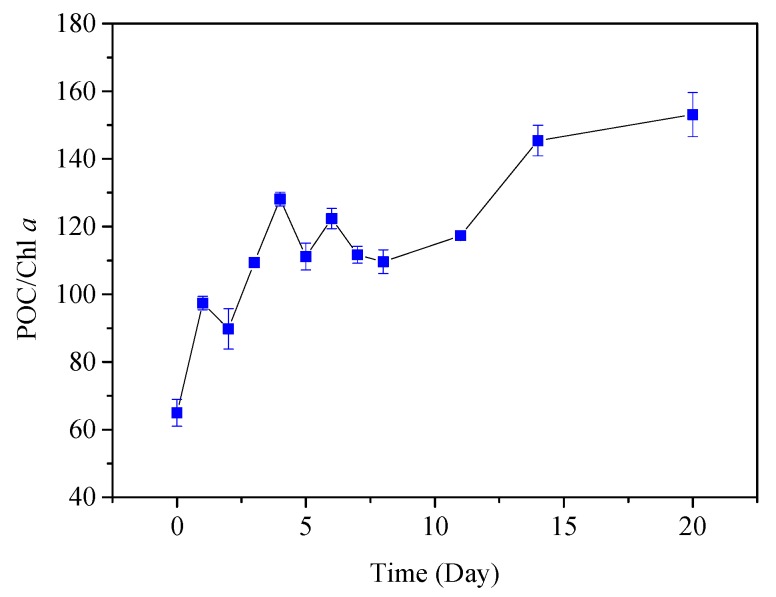
Time-series of POC: Chl *a* during cyanobacteria degradation to characterize the proportion of fresh cyanobacteria and cyanobacteria degradation residues in PPOM.

**Table 1 ijerph-16-02155-t001:** Physical and chemical indicators of water samples during PPOM degradation experiment.

Date (d)	T (°C)	Turbidity (NTU)	DO (mg·L^−1^)	pH	SS mg·L^−1^	POM mg·L^−1^	POC mg·L^−1^	POP mg·L^−1^	TPP mg·L^−1^	TP mg·L^−1^	Chl *a* μg·L^−1^
0	29.9 ± 0.1	291.65 ± 10.33	7.12 ± 0.50	7.86 ± 0.06	135.32 ± 12.38	130.12 ± 2.17	56.43 ± 2.18	0.97 ± 0.35	0.99 ± 0.34	1.12 ± 0.26	837.45 ± 112.38
2	30.1 ± 0.1	156.62 ± 10.68	3.42 ± 0.61	7.60 ± 0.09	117.65 ± 11.31	127.54 ± 13.43	74.05 ± 5.33	0.56 ± 0.11	0.59 ± 0.10	0.65 ± 0.17	740.22 ± 94.03
6	30.2 ± 0.1	108.81 ± 8.32	5.75 ± 0.62	7.58 ± 0.08	100.89 ± 18.08	89.33 ± 7.79	57.26 ± 3.49	0.53 ± 0.21	0.54 ± 0.21	0.59 ± 0.16	461.53 ± 58.57
11	30.3 ± 0.1	197.42 ± 18.87	5.94 ± 0.34	7.51 ± 0.04	115.52 ± 12.12	114.46 ± 15.78	60.38 ± 3.98	0.57 ± 0.18	0.58 ± 0.17	0.63 ± 0.22	470.65 ± 43.31
14	30.2 ± 0.1	230.07 ± 25.34	6.15 ± 0.27	7.72 ± 0.12	134.26 ± 24.56	123.35 ± 20.02	61.47 ± 5.45	0.52 ± 0.12	0.53 ± 0.12	0.66 ± 0.15	412.28 ± 83.02
20	30.5 ± 0.1	332.95 ± 20.27	6.48 ± 0.18	7.85 ± 0.10	142.75 ± 19.53	118.85 ± 20.66	57.57 ± 9.25	0.49 ± 0.28	0.50 ± 0.28	0.62 ± 0.15	417.45 ± 64.39

Note: The numbers on the left of “±” represent the mean value, and the numbers on the right of “±” represent the standard deviation.

**Table 2 ijerph-16-02155-t002:** Percentage of each P species in PPOM during PPOM degradation experiment.

Species	0d %	1d %	2d %	4d %	6d %	8d %	11d %	14d %	20d %
Ortho-P	46.68	25.63	29.00	33.40	46.04	48.24	42.83	25.71	32.14
Mono-P	38.75	51.96	48.35	46.48	45.2	45.90	34.43	37.62	37.26
Diester-P	6.66	8.29	7.90	10.28	7.09	0.00	0.81	5.27	2.38
Pyro-P	6.53	12.63	12.92	9.85	1.65	5.86	20.32	27.55	23.43
Poly-P	1.37	1.48	1.84	0.00	0.00	0.00	1.61	3.85	4.79
Pi	54.58	39.74	43.76	43.25	47.69	54.10	64.76	57.11	60.36
Po	45.41	60.25	56.25	56.76	52.31	45.90	35.24	42.89	39.64
Bio-P	53.31	74.36	71.01	66.61	53.96	51.76	57.17	74.29	67.86

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
