# Peer review of "Degradation Characteristics of Phosphorus in Phytoplankton-Derived Particulate Organic Matter and Its Effects on the Growth of Phosphorus-Deficient Microcystis aeruginosa in Lake Taihu"

_ijerph, 2019, doi:10.3390/ijerph16122155_

Round 1
Reviewer 1 Report
The manuscript by Kong et al. examines the release of phosphorus from particulate organic matter (POM) in an example lake, that has long been studied with regards to particulate P dynamics, especially suspended particulate matter (e.g., Fan et al., Science in China Series D: Earth Sciences 47.8 (2004): 710-719). The study appears to build on previous work by the same authors (http://dx.doi.org/10.3390/ijerph15112355) and it is surprising that this data and its analysis was not included in their recent publication as it appears to be (i) collected during the same collection period, (ii) uses the same samples, and (iii) is part of the previous study’s experimental design. In fact, after reading the authors’ previous work, many of their claims (regarding phyto-plankton derived POM) in the previous paper are better supported by the results in this manuscript. The result of the authors splitting these manuscripts is that, in my opinion, this secondary manuscript is somewhat weakened. In spite of this weakness, I do believe that the dataset and analyses are worthy of publication in a special issue format once the below concerns are addressed. The chemical methods are generally clear and the justification for the P data collection and analysis is also generally clear.
A) But, why the focus on Microcystis aeruginosa? This is not detailed in the introduction at all. In fact, M. aeruginosa isn’t even mentioned until the methods. As a result, currently, there is no justification/no reason for assessing M. aeruginosa growth. Thus, currently, there is no reason for half the results. I assume the focus on these particular cyanobacteria is because of its nasty neurotoxic blooms.
To continue this point: Why was the study done in such a way that M. aeruginosa was “phosphorus-deficient”? Are these cyanobacteria typically as P-deficient in Lake Taihu (or other lakes) as the authors have made them by their methods (see lines 95-98)? Was this extreme P-deficiency done to ensure consistency of inoculum between the different P sources?
B) The English language is awkward or grammatically incorrect throughout. I recommend substantial English editing. Another note on language: M. aeruginosa is technically cyanobacteria--not an algae.
C) Methods/data representation:
Line 82 – “There were three parallel experiments conducted in each group.” What groups? The group sampled on each “incubation day” for each “P source)? So, there were 3 buckets each for day 0’s G1-G5, i.e., there were 15 buckets for each day 0, 1, etc?
Line 90 – Why were the freeze-dried and ground samples “passed through 63 um sieves?” Was the portion that did not pass removed? Or, was the sample then reground until passing through the sieve? Removing sample would remove P, so clarification here is necessary.
Other nutrient limitations? P alone will not support growth. Were the cyanobacteria released from other nutrient limitations, C and N for example, to observed how well PPOM was biodegraded? If not, the increases in P and P-species may be related to mortality of rather than biodegradation. The authors do mention "cell lysis" in line 280, but a connection to the methodology is needed.
Nearly all table and figure captions provide too little explanation. Figure 1, for example, simply says “Sampling point location.” By itself, the figure and its caption convey nothing specific about the study or its reason for using this “point location” or what was sampled, etc. For other more detailed figures and tables, greater details are even more necessary. For example, Figure 2 simply states “13P NMR Spectra of PPOM,” but this provides no information about the “Days” for each spectra, what the “chemical shift” is, what the height of the “chemical shift” means, etc.
Finally, the authors' conclusions are a bit over-reaching by connecting what occurred in their controlled buckets with an inoculum of a single cyanobacterial species to a statement "that it is necessary to pay great attention to the effects of P release from PPOM degradation in order to effectively control eutrophication in the future."
Author Response
Response to Reviewer 1 Comments
Point 1: But, why the focus on Microcystis aeruginosa? This is not detailed in the introduction at all. In fact, M. aeruginosa isn’t even mentioned until the methods. As a result, currently, there is no justification/no reason for assessing M. aeruginosa growth. Thus, currently, there is no reason for half the results. I assume the focus on these particular cyanobacteria is because of its nasty neurotoxic blooms.
Response 1: Thanks for your comments and suggestions. This has been revised according to your suggestion. We added the description of Microcystis aeruginosa in the introduction. Please refer to the revised manuscript.
Point 2: To continue this point: Why was the study done in such a way that M. aeruginosa was “phosphorus-deficient”? Are these cyanobacteria typically as P-deficient in Lake Taihu (or other lakes) as the authors have made them by their methods (see lines 95-98)? Was this extreme P-deficiency done to?
Response 2: Thanks for your comments and suggestions. Various species of P in water body has different nutritional supply ability to the growth of M. aeruginosa. Soluble reactive P (SRP) in water body can be directly utilized by M. aeruginosa. Under the environmental conditions of "Phosphorus Deficiency", M. aeruginosa can hydrolyze organic P into inorganic P by releasing phosphatase to maintain its growth. Therefore, the migration and transformation of P in PPOM may play an important role in the growth of Microcystis. According to the literature” XU Hai, WU Yali, YANG Guijun, et al. Tolerance of Microcystis aeruginosa and Scendesmus obliquus to nitrogen and phosphorus deficiency. Ecological Science, 2014, 33(5): 879-884 (In Chinese).”, M. aeruginosa could maintain exponential growth for about eight days under phosphorus deficiency condition. In order to consume the phosphorus absorbed by M. aeruginosa thoroughly, we prolonged the incubation time (15 day) appropriately compared to Xu et al,2014 (8 day).
Point 3. The English language is awkward or grammatically incorrect throughout. I recommend substantial English editing. Another note on language: M. aeruginosa is technically cyanobacteria--not an algae.
Response 3: Thanks for your comments and suggestions. English throughout the manuscript have been carefully examined and revised according to your suggestion. Please refer to the revised manuscript. Algae has been replaced by cyanobacteria when describe M. aeruginosa.
Point 4. Line 82 – “There were three parallel experiments conducted in each group.” What groups? The group sampled on each “incubation day” for each “P source)? So, there were 3 buckets each for day 0’s G1-G5, i.e., there were 15 buckets for each day 0, 1, etc?
Response 4: Thanks for your comments and suggestions. The sentence “There were three parallel experiments conducted in each group.” has been revised to “Three parallel groups were set in the experiment.”. This means three plastic buckets (80 L) were used in the simulation experiment. Please refer to the revised manuscript.
Point 5. Line 90 – Why were the freeze-dried and ground samples “passed through 63 um sieves?” Was the portion that did not pass removed? Or, was the sample then reground until passing through the sieve? Removing sample would remove P, so clarification here is necessary.
Response 5: Thanks for your comments and suggestions. This has been revised according to your suggestion. The purpose of samples passed through 63 um sieves is to make sure the digestion process is complete and the measurement results are more accurate when the total phosphorus and organic carbon are measured. And all the samples passed through the sieve avoid loss of quality. Please refer to the revised manuscript.
Point 6. Other nutrient limitations? P alone will not support growth. Were the cyanobacteria released from other nutrient limitations, C and N for example, to observed how well PPOM was biodegraded? If not, the increases in P and P-species may be related to mortality of rather than biodegradation. The authors do mention "cell lysis" in line 280, but a connection to the methodology is needed.
Response 6: Thanks for your comments and suggestions. We agree with your opinion that P alone will not support growth. It is generally believed that phosphorus is the main limiting factor for cyanobacteria growth in aquatic ecosystems. According to the ratio of Redfield, the ratio of C, N and P in cyanobacteria cells is about 106:16:1 (Redfield A C. 1963. The influence of organisms on the composition of sea-water. Sea, 40: 640, 644). So P is often the limiting factor for cyanobacteria growth. According to the results of this study, cyanobacteria density showed an upward trend or maintained a relatively stable level (Fig.4), which indicated that no obvious death of algae occurred. We speculate that P in the system should not come from the phosphorus released by algae death, but mainly from the degradation of particulate matter. We also completely agree with your suggestion. In the future research, it is necessary to carry out the experimental observation of algae-derived particulate matter degradation.
Point 7. Nearly all table and figure captions provide too little explanation. Figure 1, for example, simply says “Sampling point location.” By itself, the figure and its caption convey nothing specific about the study or its reason for using this “point location” or what was sampled, etc. For other more detailed figures and tables, greater details are even more necessary. For example, Figure 2 simply states “13P NMR Spectra of PPOM,” but this provides no information about the “Days” for each spectra, what the “chemical shift” is, what the height of the “chemical shift” means, etc.
Response 7: Thanks for your comments and suggestions. This has been revised according to your suggestion. Detailed descriptions and explanations of graphs and tables in titles and articles are given. Please refer to the revised manuscript.
Point 8. Finally, the authors' conclusions are a bit over-reaching by connecting what occurred in their controlled buckets with an inoculum of a single cyanobacterial species to a statement "that it is necessary to pay great attention to the effects of P release from PPOM degradation in order to effectively control eutrophication in the future."
Response 8: Thanks for your comments and suggestions. We agree with your opinion, this has been revised according to your suggestion. Please refer to the revised manuscript.

Reviewer 2 Report
Materials and method
(1) L 80, the description of the simulation experiment set is not clear.
(2) L 99, culture temperature was 26 oC, have the authors measured the temperature during the experiment?
(3) L149, How do the authors compare the data difference at different time and/or in different groups? The statistical analysis is not sufficient.
Results:
(4) L170, in the tables and figures, the significant difference of the data would be displayed by symbol.
(5) L 189 “n.d.” explanation, but there is no such symbol in the table.
Discussion:
(6) L237, “studies” but only one ref.
(7) Have the authors extended the experiment to find the POC:Chl a more than 200? When will it last the degradation of Lake Taihu water bloom?
Author Response
Response to Reviewer 2 Comments
Point 1. L 80, the description of the simulation experiment set is not clear.
Response 1: Thanks for your comments and suggestions. This has been revised according to your suggestion. Please refer to the revised manuscript.
Point 2. L 99, culture temperature was 26 oC, have the authors measured the temperature during the experiment?
Response 2: Thanks for your comments and suggestions. The constant temperature incubator was set at 25±1℃, This has been revised in the manuscript. Please refer to the revised manuscript.
Point 3. L149, How do the authors compare the data difference at different time and/or in different groups? The statistical analysis is not sufficient
Response 3: Thanks for your comments and suggestions. SPSS software is mainly used to analyze the difference of data. This has been revised in the manuscript. Please refer to the revised manuscript.
Point 4. L170, in the tables and figures, the significant difference of the data would be displayed by symbol.
Response 4: Thanks for your comments and suggestions. This has been revised according to your suggestion. Please refer to the revised manuscript.
Point 5. L 189 “n.d.” explanation, but there is no such symbol in the table.
Response 5: Thanks for your comments and suggestions. This has been revised according to your suggestion. Please refer to the revised manuscript.
Point 6. L237, “studies” but only one ref.
Response 6: Thanks for your comments and suggestions. This has been revised according to your suggestion. Please refer to the revised manuscript.
Point 7. Have the authors extended the experiment to find the POC:Chl a more than 200? When will it last the of Lake Taihu water bloom?
Response 7: Thanks for your comments and suggestions. During the experimental period, we found that the ratio of POC:Chl a was less than 200. In future studies, we will extend the experimental time to observe the change of the POC:Chl a ratio and the time of algal blooms degradation.

Reviewer 3 Report
Review
The manuscript entitled “Degradation characteristics of phosphorus in phytoplankton-derived particulate organic matter and its effects on the growth of phosphorus-deficient Microcystis aeruginosa in Lake Taihu” describes Particulate Organic Matter characteristics, its degradation, release of P and its effects on the growth of P-deficient coccoid cyanobacterium studied during the experiments.
I find this manuscript interesting, however, I have some suggestions and questions. For example, lines 55-57 – it seems that this is the summary of your results – if not, and this part follows the literature described above, your study brings nothing new. You should highlight novelty of your work, like in the line 60. Line 63 – I think that the effects of PPOM on development of M. aeruginosa is also very important and interesting part of your work – it is missing in purposes.
What is more important, results are not clearly presented and described. I found some divergences that must be clarified.
English language is correct.
Other comments:
Line 14 – change “algal” to “cyanobacterial”
Lines 17, 229, 244, 278, 313 – change “ algae” to “phytoplankton”. Algae is more often used to describe eukaryotic photosynthetic microorganisms whereas in “phytoplankton” term we have also cyanobacteria included as phytoplankton is some kind of artificial term describing all planktonic, mostly photosynthetic prokaryotes and eukaryotes.
Line 18 – did you measure TP concentration? I can not find this information. Or maybe you mean TPP? Clarify, please.
Lines 18-19 – Is this true? Is this based on results described in Table 2? If yes, it’s not true as pyro-P was not degraded completely, whereas Diester-P was degraded after 4 days. Can you check it and explain and describe results carefully in the manuscript?
Line 25 – change “P” to “phosphorus”
Line 65 – add “water of” before “shallow eutrophic…” and delete “water” after “lake”
Line 84 – delete “A” and change “was” to “were”
Line 89 – use one unit and change “0.064 mm” to “64 µm”
Lines 22, 94 – use a shortcut of the species name “M. aeruginosa” , as it was used earlier in the text. Rather “obtained” than “isolated”
Line 96 – “were centrifuged.. “ give the speed and time.
2.2.2 – add in this section information about the exposure time, for example in line 100.
Line 155 – change “is” to “was”
Line 207 – add “a” and “b” in the figure caption
Lines 214-215 – “which was slightly lower…”– on the basis of standard deviation bars - it was rather the same. It seems that there was no statistically significant differences especially between G1 and G5 and between G3 and G5. Rewrite this sentence, please.
Lines 249-251 – You describe here results, that were not presented earlier. It’s confusing. I can not find these results in the text, tables or figures. Line 253 – “Diester-P decreased to 0 on the 4th day..” whereas in the Table 2 (this is the only table describing results of PPOM degradation in detail) Diester-P decreased to 0 on the 8th day… Can you explain this? The results should be described carefully. Change the title of the table for example ” Percentage and content of each P species in POM during degradation experiment” , the same with table 1 – are these values from experiment on PPOM degradation or M. aeruginosa exposure to PPOM? Clarify, please in the table caption.
Line 251 – remove “adsorbed”
Line 274 – Figure 3 not 4.
Lines 258-264 –you should specify the term “algae” here, and change it to “phytoplankton”, “cyanobacteria” or “M. aeruginosa”
Line 269 – delete dot after “lakes”
Line 292 – change to “cyanobacterial growth”
Line 296 – change “studies” to “study” as you cite only one literature
Line 313 – why after 48 h? I do not understand this conclusion: why ortho-P was released since it was present at time 0h? – just at the start of the experiment.
Lines 381, 414, – add a dot after page number
Line 460 – delete a dot after page number
Author Response
Response to Reviewer 3 Comments
Point 1. I find this manuscript interesting, however, I have some suggestions and questions. For example, lines 55-57 – it seems that this is the summary of your results – if not, and this part follows the literature described above, your study brings nothing new. You should highlight novelty of your work, like in the line 60. Line 63 – I think that the effects of PPOM on development of M. aeruginosa is also very important and interesting part of your work – it is missing in purposes.
Response 1: Thanks for your comments and suggestions. This has been revised according to your suggestion. We have rephrased Lines 55-57 and Line 63 according to your suggestion. Please refer to the revised manuscript.
Point 2. What is more important, results are not clearly presented and described. I found some divergences that must be clarified.
Response 2: Thanks for your comments and suggestions. We have rephrased the results according to your suggestion. Please refer to the revised manuscript.
Point 3. Line 14 – change “algal” to “cyanobacterial”
Response 3: Thanks for your comments and suggestions. This has been revised according to your suggestion. Please refer to the revised manuscript.
Point 4. Lines 17, 229, 244, 278, 313 – change “ algae” to “phytoplankton”. Algae is more often used to describe eukaryotic photosynthetic microorganisms whereas in “phytoplankton” term we have also cyanobacteria included as phytoplankton is some kind of artificial term describing all planktonic, mostly photosynthetic prokaryotes and eukaryotes.
Response 4: Thanks for your comments and suggestions. This has been revised according to your suggestion. Please refer to the revised manuscript.
Point 5. Line 18 – did you measure TP concentration? I can not find this information. Or maybe you mean TPP? Clarify, please.
Response 5: Thanks for your comments and suggestions. We have measured TP concentration and data on TP have been supplemented. Sorry for missing TP data in the manuscript. This has been revised according to your suggestion. Please refer to the revised manuscript.
Point 6. Lines 18-19 – Is this true? Is this based on results described in Table 2? If yes, it’s not true as pyro-P was not degraded completely, whereas Diester-P was degraded after 4 days. Can you check it and explain and describe results carefully in the manuscript?
Response 6: Thanks for your comments and suggestions. This has been revised according to your suggestion. Please refer to the revised manuscript.
Point 7. Line 25 – change “P” to “phosphorus”
Response 7: Thanks for your comments and suggestions. This has been revised according to your suggestion. Please refer to the revised manuscript.
Point 8. Line 65 – add “water of” before “shallow eutrophic…” and delete “water” after “lake”;
Response 8: Thanks for your comments and suggestions. Table 2 has been re-structured according to your suggestion. Please refer to the revised manuscript.
Point 9. Line 84 – delete “A” and change “was” to “were”.
Response 9: Thanks for your comments and suggestions. We have rephrased the conclusions according to your suggestion. Please refer to the revised manuscript.
Point 10. se one unit and change “0.064 mm” to “64 µm”
Response 10: Thanks for your comments and suggestions. The sentences between rows 63-71 have been simplify according to your suggestion. Please refer to the revised manuscript.
Point 11. Lines 22, 94 – use a shortcut of the species name “M. aeruginosa” , as it was used earlier in the text. Rather “obtained” than “isolated”
Response 11: Thanks for your comments and suggestions. All the above mistakes have been corrected according to your suggestion. Please refer to the revised manuscript.
Point 12. Line 96 – “were centrifuged.. “ give the speed and time.
Response 12: Thanks for your comments and suggestions. This has been revised according to your suggestion. Please refer to the revised manuscript.
Point 13. 2.2.2 – add in this section information about the exposure time, for example in line 100.
Response 13: Thanks for your comments and suggestions. This has been revised according to your suggestion. Please refer to the revised manuscript.
Point 14. Line 155 – change “is” to “was”
Response 14: Thanks for your comments and suggestions. This has been revised according to your suggestion. Please refer to the revised manuscript.
Point 15. Line 207 – add “a” and “b” in the figure caption
Response 15: Thanks for your comments and suggestions. This has been revised according to your suggestion. Please refer to the revised manuscript.
Point 16. Lines 214-215 – “which was slightly lower…”– on the basis of standard deviation bars - it was rather the same. It seems that there was no statistically significant differences especially between G1 and G5 and between G3 and G5. Rewrite this sentence, please.
Response 16: Thanks for your comments and suggestions. This has been revised according to your suggestion. Please refer to the revised manuscript.
Point 17. Lines 249-251 – You describe here results, that were not presented earlier. It’s confusing. I can not find these results in the text, tables or figures. Line 253 – “Diester-P decreased to 0 on the 4thday..” whereas in the Table 2 (this is the only table describing results of PPOM degradation in detail) Diester-P decreased to 0 on the 8th day… Can you explain this? The results should be described carefully. Change the title of the table for example ” Percentage and content of each P species in POM during degradation experiment” , the same with table 1 – are these values from experiment on PPOM degradation or M. aeruginosa exposure to PPOM? Clarify, please in the table caption.
Response 17: Thanks for your comments and suggestions. This has been revised according to your suggestion. Lines 249-251 has been rewritten. Sorry for our mistake, we reconfirm that Diester-P decreased to 0 on the 8th day. The titles of Table 1 and Table 2 have been rewritten. Please refer to the revised manuscript.
Point 18. Line 251 – remove “adsorbed”
Response 18: Thanks for your comments and suggestions. This has been revised according to your suggestion. Please refer to the revised manuscript.
Point 19. Line 274 – Figure 3 not 4.
Response 19: Thanks for your comments and suggestions. This has been revised according to your suggestion. Please refer to the revised manuscript.
Point20. Lines 258-264 –you should specify the term “algae” here, and change it to “phytoplankton”, “cyanobacteria” or “M. aeruginosa”
Response 20: Thanks for your comments and suggestions. This has been revised according to your suggestion. Please refer to the revised manuscript.
Point 21. Line 269 – delete dot after “lakes”
Response 21: Thanks for your comments and suggestions. This has been revised according to your suggestion. Please refer to the revised manuscript.
Point 22. Line 292 – change to “cyanobacterial growth”
Response 22: Thanks for your comments and suggestions. This has been revised according to your suggestion. Please refer to the revised manuscript.
Point 23. Line 296 – change “studies” to “study” as you cite only one literature
Response 23: Thanks for your comments and suggestions. This has been revised according to your suggestion. Please refer to the revised manuscript.
Point 24. Line 313 – why after 48 h? I do not understand this conclusion: why ortho-P was released since it was present at time 0h? – just at the start of the experiment.
Response 24: Thanks for your comments and suggestions. This has been revised according to your suggestion. Please refer to the revised manuscript.
Point 25. Lines 381, 414, – add a dot after page number
Response 25: Thanks for your comments and suggestions. This has been revised according to your suggestion. Please refer to the revised manuscript.
Point 26. Line 460 – delete a dot after page number
Response 26: Thanks for your comments and suggestions. This has been revised according to your suggestion. Please refer to the revised manuscript.

Round 2
Reviewer 1 Report
The revised manuscript satisfied all but one of my concerns. The study overall is well done and, in my opinion, ready for publication after some minor edits to the table and figure captions. I appreciate that the authors have already improved the caption text, but the captions still omit critical information. Specifically:
Figure 1 is ok
Figure 2 - Please define the y-axis (what are the peak units for the NMR spectra?).
Figure 3 - Please define acronyms on x- and y-axes (APA, EHP, TPP) in the caption, or do not use acronyms for axis labels. (two minor edits: (1) why do the authors switch from standard unit notation, mg L-1, to fraction unit notation, mg/L, in this figure? Can the unit notation be changed to standard? (2) why are the units different (mmol L-1 verus mg L-1)? It would be nice if all variables, where possible, were the same units for comparability.
Figures 4 through 6 - Please define G1-G5 in the legend or caption. Remember that figures and tables should, to some degree, be able to stand alone. Also, again, please change unit notation to standard (i.e., mL-1)
Figure 7 - This caption should very briefly state the purpose of plotting POC:Chl a over time.
Table 1 - what are the values presented? Are they mean, median or something else? What is the +/- value? variance, standard deviation, standard error, or something else?
Table 2 - again, what are the values presented? Mean, median, the magnitude of a single observation? In this figure, if these are estimates of central tendency, please provide estimates of error/variability like provided in Table 1.
With these edits for clarity, I recommend publication.
Author Response
Point 1: Figure 2 - Please define the y-axis (what are the peak units for the NMR spectra?).
Response 1: Thanks for your comments and suggestions. The y-axis represents the peak strength. It's a relative value. In general, no vertical coordinates are shown on the spectrum. For example, Bai et al., Classes of dissolved and particulate phosphorus compounds and their spatial distributions in the water of a eutrophic lake: a 31P NMR study, Biogeochemistry (2015) 126:227–240; Liu et al. Characterization of plant-derived carbon and phosphorus in lakes by sequential fractionation and NMR spectroscopy, Science of the Total Environment 566–567 (2016) 1398–1409. Please refer to the revised manuscript.
Point 2: Figure 3 - Please define acronyms on x- and y-axes (APA, EHP, TPP) in the caption, or do not use acronyms for axis labels. (two minor edits: (1) why do the authors switch from standard unit notation, mg L-1, to fraction unit notation, mg/L, in this figure? Can the unit notation be changed to standard? (2) why are the units different (mmol L-1 verus mg L-1)? It would be nice if all variables, where possible, were the same units for comparability.
Response 2: Thanks for your comments and suggestions. This has been revised according to your suggestion. Acronyms of APA, EHP and TPP have been defined on x- and y-axes in the caption. The unit has been modified to mg·L-1, Alkaline phosphatase activity (APA) was determined with p-NPP as the reaction substrate, p-NPP was a mixture, which was more convenient and accurate by using mmol L-1. Please refer to the revised manuscript.
Point 3: Figures 4 through 6 - Please define G1-G5 in the legend or caption. Remember that figures and tables should, to some degree, be able to stand alone. Also, again, please change unit notation to standard (i.e., mL-1)
Response 3: Thanks for your comments and suggestions. This has been revised according to your suggestion. Please refer to the revised manuscript.
Point 4: Figure 7 - This caption should very briefly state the purpose of plotting POC:Chl a over time.
Response 4: Thanks for your comments and suggestions. This has been revised according to your suggestion. Please refer to the revised manuscript.
Point 5: Table 1 - what are the values presented? Are they mean, median or something else? What is the +/- value? variance, standard deviation, standard error, or something else?
Response 5: Thanks for your comments and suggestions. The numbers on the left of “±” represent the mean value, and the numbers on the right of “±” represent the standard deviation. We made notes at the bottom of the table,. Please refer to the revised manuscript.
Point 6: Table 2 - again, what are the values presented? Mean, median, the magnitude of a single observation? In this figure, if these are estimates of central tendency, please provide estimates of error/variability like provided in Table 1.
Response 6: Thanks for your comments and suggestions. This has been revised according to your suggestion. The values as a single observation which represented the percentage of each P species in PPOM during PPOM degradation experiment (%), the units were added in the table. Please refer to the revised manuscript.

Reviewer 3 Report
Review
I recommend to accept after minor revision.
Line 20 – start as a new sentence “The variation of…”
Line 63 – This is not entirely true. Maybe change to “ Microcystis is one of the dominant genera in phytoplankton of eutrophic lakes”
Line 90 – delete “in”
Conclusion – check and correct two last sentences, please.
Author Response
Point 1: Line 20 – start as a new sentence “The variation of…”
Response 1: Thanks for your comments and suggestions. This has been revised according to your suggestion. Please refer to the revised manuscript.
Point 2: Line 63 – This is not entirely true. Maybe change to “ Microcystis is one of the dominant genera in phytoplankton of eutrophic lakes”
Response 1: Thanks for your comments and suggestions. This has been revised according to your suggestion. Please refer to the revised manuscript.
Point 3: Line 90 – delete “in”
Response 1: Thanks for your comments and suggestions. This has been revised according to your suggestion. Please refer to the revised manuscript.
Point 4: Conclusion – check and correct two last sentences, please.
Response 1: Thanks for your comments and suggestions. This has been revised according to your suggestion. Please refer to the revised manuscript.
